# Modeling of Satellite Constellation in Modelica and a PHM System Framework Driven by Model Data Hybrid

**Chan Liu \*** , **Liping Chen, Jianwan Ding and Duansen Shangguan**

Department of Mechanical Science and Engineering, Huazhong University of Science and Technology, Wuhan 430074, China; chenlp@hust.edu.cn (L.C.); dingjw@hust.edu.cn (J.D.); ahcq1990@hust.edu.cn (D.S.)
\* Correspondence: liuchan@hust.edu.cn

**Abstract:** The new generation of low-earth-orbit (LEO) satellite constellation systems has the characteristics of low delay, strong signal and global coverage, and it is an important direction for the development of next-generation communication technology. A major disadvantage is that the constellation system is huge, often composed of hundreds or thousands of satellites, which puts forward high requirements for the design and health management of the constellation system, and the existing telemetry data monitoring system cannot meet the actual needs. CPS is a multidimensional complex system that integrates computation, communication and control (3C). Through the deep integration and cooperation of 3C, the real-time monitoring and dynamic control of large-scale engineering systems are realized, which is completely suitable for the operation and maintenance requirements of the satellite constellation system. This paper firstly establishes the entire satellite constellation system model, which is integrated from the satellite multidomain system, the constellation orbit environment system and the communication link system. Then, according to the technical concept of cyber-physical systems (CPS), an implementation framework of a prognostics and health (PHM) system driven by a model–data hybrid for satellite constellation systems is proposed. The framework is based on model simulation data and telemetry data and combines virtual and real data fusion, fault diagnosis, simulation prediction and other technologies to generate enhanced data to drive the effective operation of the PHM system. Finally, a verification case is designed to prove that the satellite constellation health management system implemented under this framework has a positive effect on the reliable operation and maintenance of the satellite constellation system.

**Keywords:** cyber-physical systems; Modelica; modeling and simulation; model-based systems; satellite constellation; model–data hybrid

## 1. Introduction

In recent years, with the exponential increase in the demand for terrestrial network communication, LEO-satellite-constellation-based communication systems have attracted great attention due to their low time delay and region-wide coverage. The idea of providing the Internet from space has created a global boom. A number of LEO communication satellite constellations have emerged internationally, which are represented by OneWeb, StarLink, Telesat and Iridium NEXT. With the maturation and development of microsatellite manufacturing technology and one-arrow multistar technology, the emerging constellations are usually characterized by small-sized structures, giant constellations (100 to 12,000), short construction periods and high investment costs [1]. Therefore, PHM systems are under great pressure. The PHM system of the satellite constellation monitors, adjusts and controls the operation process of the constellation. It is the core of the management of global resources and system status. It directly determines whether the constellation can operate well and meet the communication needs of users [2].

Traditional PHM systems for satellites are mostly driven by pure data. The telemetry data from the space–ground link is displayed on the monitoring interface to operation

engineers. When one or more telemetry parameters exceed the predefined safety threshold, the operation engineer will try to locate the fault with the aid of a fault diagnosis algorithm [3]. Obviously, the traditional PHM method for satellites is not suitable for satellite constellation systems that have higher requirements on reliability for three reasons. Firstly, there is a lag in data-driven health monitoring technology because many faults do not occur suddenly, the abnormal behavior caused by the early degradation of spacecraft components is very subtle and it is easy for technicians to ignore when they simply use telemetry data for monitoring. In the form of a threshold value, it indicates that the fault was not detected in time, and it is likely that the relevant components have been damaged [4]. Second, when a satellite enters nonmeasuring and control areas and loses telemetry data, data-driven methods are not applicable [5]. Third, the variables contained in telemetry data may be less sensitive to fault characteristics, which will reduce the efficiency and accuracy of the fault detection algorithm [6]. In this context, introducing a virtual model of satellite constellation to describe the behavior of constellation entities and combining virtual models with telemetry data is an important way to make up for the shortage of telemetry data [7].

So far, the modeling of the constellation has been developed in three main ways, namely, the model of the satellite multidomain system, the model of the constellation system and the model of the communication link system. The satellite system is a typical complex system with mechanical, electrical, thermal and control coupling and the modeling method based on the Modelica language has made great achievements in this respect. As early as 2009, Zhang Huijing established the Modelica model of the satellite flywheel, which verified the feasibility of Modelica language in satellite multidomain modeling [8]. In 2017, Jonathan Shum explored the technical roadmap of integrating satellite dynamics and control models developed in the Modelica language into the mechanical design cycle of satellites with certain success [9]. Later, Liu Yuhui et al. performed modeling on the whole satellite system and studied the effects of the coupling of the dynamics subsystem, attitude control subsystem, propulsion subsystem and energy subsystem on the change in satellite status [10]. The mainstream software used in the construction of the constellation system model includes FreeFlyer, STK and Savi, which allow the computation and visualization of the orbit, motion and ground coverage of the satellite to simulate constellation orbits and space environments [11]. In 2021, Wang Qian [12] used STK software to model a constellation of GPS M-code satellites. Through simulation, the visible number of satellites and the distribution of the geometric accuracy factors of the constellation in different regions of the world are obtained, and the constellation is verified to be able to provide stable and reliable communication services around the world. The communication link system model mainly consists of network simulation software suitable for constellation simulation environments, such as OPNET Modeler, ns-3, etc., which adopts discrete event simulation technology and provides a variety of network protocols and functions to simulate the communication processes, such as satellite paging, chain building and link switching, during high-speed motion and to evaluate the communication quality. In 2018, Hu Jinhua built a low-orbit satellite constellation mobile communication system simulation platform based on OPNET which can correctly simulate the main communication processes of LEO communication systems and the characteristics of satellite communication such as long-time delay and frequent switching [13].

In a real constellation, both the operational state of the satellite itself and the orbital distribution and configuration of the constellation system as a whole affect the quality of the communication links, but almost all published papers on the constellation consider these three aspects separately. Mostly, STK orbital data are imported into OPNET for one-way interaction. Moreover, there is very limited work in Modelica on the modeling of the constellation system and communication links as well. In this paper, a satellite constellation system model is developed in MWorks, and an attempt is made to couple the satellite multidomain system model, constellation system model and communication link model into a system model to achieve a multiscenario simulation of the satellite constellation system under a unified platform. Then, channels are established for an interaction between

telemetry data and the Modelica model for the model to follow the state changes of the constellation entity, providing a new method for a more efficient PHM of constellations.

The main innovations in this paper include: (1) the development of a common model library for satellite constellation simulation based on MWorks which supports a multidomain simulation of a satellite system, constellation orbit, configuration and ground cover simulation and communication link simulation, providing a new solution for the simulation of satellite constellations; (2) a PHM system architecture driven by a model–data hybrid being proposed for the satellite constellation which compensates for the inadequacies of traditional data-driven PHM and provides a new idea for more efficient and reliable operation and maintenance (O&M) of constellation systems.

This paper's remaining part is structured as follows: Section 2 introduces the modeling theory, model architecture and implementation of the satellite constellation system. Section 3 presents the architecture of the satellite constellation PHM system based on a model–data hybrid and describes the key technologies. After that, Section 4 introduces the application results of this framework in a single constellation. Finally, Section 7 summarizes the entire paper.

## 2. Unified Modeling of LEO Satellite Constellation

This section proposes a unified model architecture for satellite multidomain simulation, constellation system simulation and communication link simulation. It also describes in detail the principles and implementation of the important components of the LEO satellite constellation system model developed by MWorks/Modelica.

### 2.1. System Principle

The LEO communication satellite constellation refers to a constellation system distributed at an orbital altitude of 200 km to 2000 km [14]. It is usually divided into three parts: space segment, ground segment and user segment, as shown in Figure 1. Among them, the space segment refers to all satellites in the constellation. The ground segment usually consists of ground stations, system control centers and integrated ground-based networks responsible for satellite monitoring and control in the space segment as well as for the management of the space network. The user segment refers to all types of user terminals, including mobile terminals, ship terminals, vehicle-mounted terminals, airborne terminals, etc.

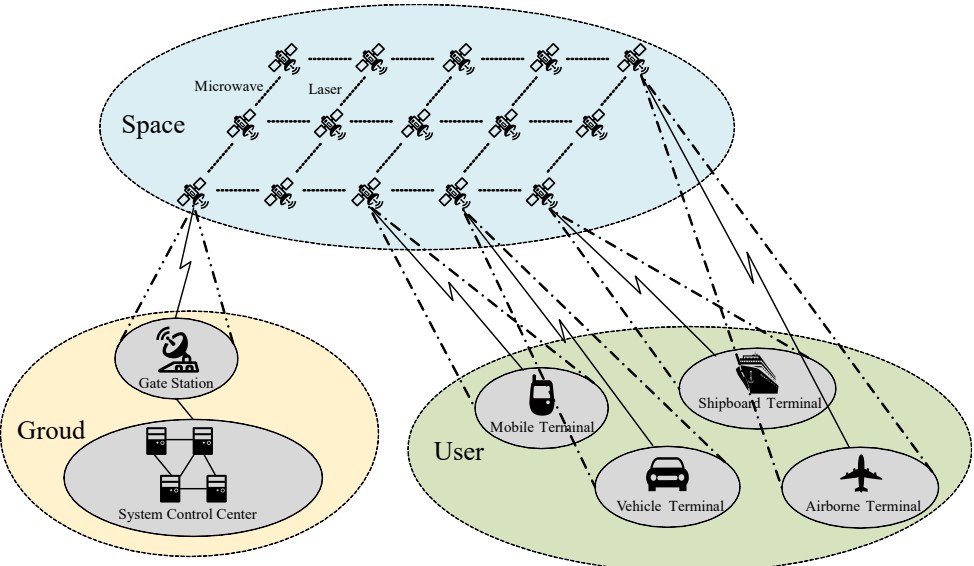

**Figure 1.** Schematic of an LEO communication satellite constellation.

The communication functions of the entire constellation system are realized in three main types of links, namely, feeder links, intersatellite links (ISL) and user links. Among them, the feeder link and the user link belong to satellite-ground links, with the difference that the feeder link connects the satellite and the ground station and the user link connects the satellite and user terminals. ISL refers to a communication link between any satellite in the constellation and a neighboring satellite in the same orbital plane or a neighboring satellite in an adjacent orbital plane. In order to cover the three links and realize satellite multidomain simulation, constellation system simulation and communication link simulation, key modeling objects are identified in Table 1.

**Table 1.** Key components of satellite constellation system modeling.

| Names | Functions |
|---|---|
| Ground station | Uplink and downlink data forwarding |
| System control center | Generating satellite maneuver command to maintain the stability of constellation configuration |
| Satellite system | The simulation of the operational status of the four subsystems of satellite energy, control, propulsion and payload |
| User terminal | Generating communication requests and sending or receiving communication data |
| Satellite–ground link | The transmission of data between satellite and ground and the calculation of time delay and communication margin |
| Intersatellite link | The transmission of data between satellites and the calculation of time delay and communication margin |
| Constellation system | The calculation of the trajectories of each satellite within the constellation according to the constellation design parameters |

*2.2. Model Architecture*

The architecture of the LEO constellation system model is shown in Figure 2, where each colored line with an arrow represents a specific type of interface [15].

- Black solid lines represent the orbital information interface for all satellites, calculated by the constellation system model based the given constellation design parameters and then transmitted to the ground station and system control center via a satellite-to-ground link.
- Red solid lines represent the command interface of the satellite maneuvers, which are generated by the system control center according to the configuration changes of the constellation to control the on-off of the satellite orbital engine and maintain the constellation configuration.
- Blue solid lines represent the energy interface, through which the electrical power generated by the solar panel is sent to other subsystems.
- Brown solid lines mean the communication request interface, which contains the communication type and the latitude and longitude information of the originating terminal and the ending terminal.
- Orange dashed lines represent the link interface, which includes information such as link margin, time delay, bandwidth ratio, etc., calculated by the link models.

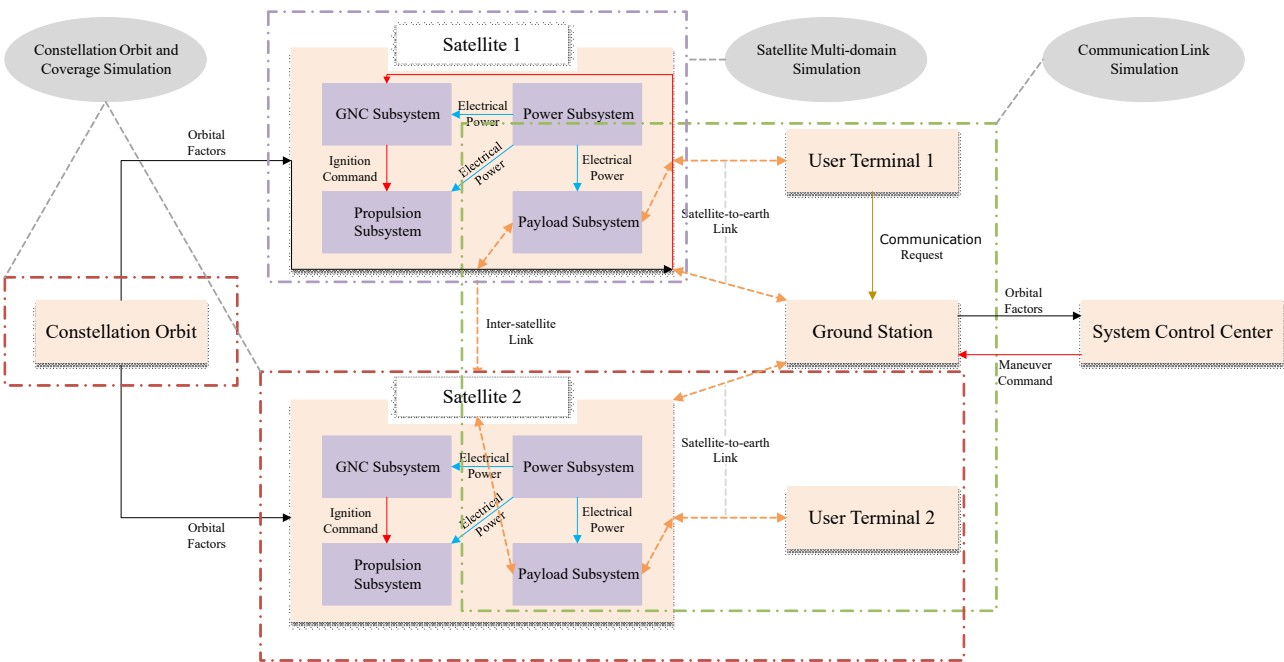

**Figure 2.** Model architecture of LEO communication satellite constellation system.

### 2.3. Model Implementation Based on Modelica

The orbit, satellite and system control center are the three most important parts of an LEO satellite constellation system, so this section mainly introduces the principles for model implementation.

### 2.3.1. Orbit Model

The main parameters of the ideal constellation orbit model are: the number of orbits in the constellation *p*, the number of satellites in each orbit plane *s* and orbital factors (semi-major axis *a*, eccentricity *e*, inclination *i*, right ascension of ascending node (RAAN) $\Omega$, argument of perigee $\omega$ and true fault *f*). The number of orbits and the number of satellites in each orbit plane determine the scale of the constellation. The semi-major axis and eccentricity of the orbit determine the size and shape of the orbit. The inclination, RAAN and argument of perigee determine the orientation and direction of the orbit. The true fault decides the position of the satellite. Based on the above parameters, the coordinates (*x,y,z*) of each satellite in the geocentric inertial system can be calculated by Equation (1) [16], so as to simulate the orbital state of the constellation at any time under ideal conditions.

$$
\begin{bmatrix} x \\ y \\ z \end{bmatrix} = \frac{a(1-e^2)}{1+e\cos f} \cdot
$$

$$
\begin{bmatrix} \cos\Omega\cos(\omega+f) - \sin\Omega\sin(\omega+f)\cos i \\ \sin\Omega\cos(\omega+f) - \cos\Omega\sin(\omega+f)\cos i \\ \sin(\omega+f)\sin i \end{bmatrix} \tag{1}
$$

In order to simulate the state of satellite orbits in space more realistically, orbital perturbations [17], including aspheric gravity, atmospheric drag, sun-moon gravity, sunlight pressure and the post-Newton effect, are introduced, as shown in Figure 3.

- Nonspherical gravitational perturbation.

The Earth's gravitational field model adopts $70 \times 70$ Joint Gravity Model 3 (JCM3). The calculation of nonspherical gravitational perturbation acceleration projected to the spherical coordinate component in the ground-fixed coordinate system is as follows [18]:

$$
\begin{aligned}
a_N = & -\frac{\mu}{r^2 \cos \varphi} \sum_{n=2}^{\infty} \sum_{m=0}^{n} (\frac{R_e}{r})^n \{(1+n) \cos \varphi \\
& \mathbf{P_{nm}}(\sin \varphi)(\overline{C}_{nm} \cos m\lambda + \overline{S}_{nm} \sin m\lambda)e_r \\
& [n \sin \varphi \mathbf{P_{nm}}(\sin \varphi) - N_n m \mathbf{P_{n-1,m}}(\sin \varphi)] \\
& (\overline{C}_{nm} \cos m\lambda + \overline{S}_{nm} \sin m\lambda)e_\varphi + m \cdot \\
& (\overline{C}_{nm} \sin m\lambda - \overline{S}_{nm} \cos m\lambda)e_\lambda \}
\end{aligned}
\tag{2}
$$

where $e_r$, $e_\varphi$ and $e_\lambda$ are the three orthogonal unit vectors of spherical coordinates. $\lambda$ and $\varphi$ denote geocentric longitude and geocentric latitude. $\overline{C}_{nm}$ and $\overline{S}_{nm}$ are the normalized gravitational coefficients. $N_{nm} = \sqrt{\frac{2n+1}{2n-1}(n+m)(n-m)}$. $\mathbf{P_{nm}}(\mathbf{u})$ is the normalized gravitational coefficient. $n$ represents the truncation order.

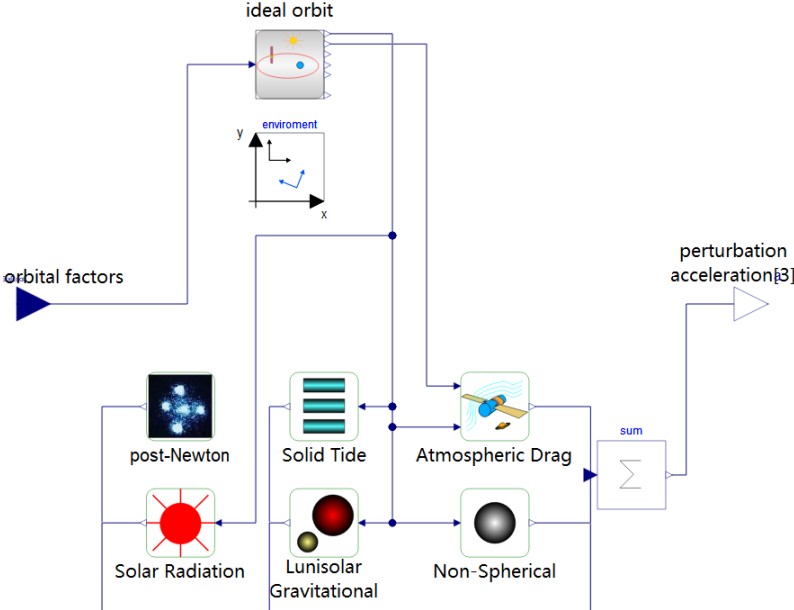

**Figure 3.** Orbit perturbation model.

- Atmospheric drag perturbation.

    The acceleration of atmospheric drag on a satellite with an area-to-mass ratio is [18]:

$$
a_D = -\frac{1}{2}C_D \frac{S}{m} \rho |V| \cdot V
\tag{3}
$$

where $C_D$ is the damping coefficient. $\rho$ is the atmospheric density at the location of the satellite. $V$ is the speed of the satellite relative to the atmosphere.

- Lunisolar gravitational perturbation.

    The perturbation acceleration of the sun and the moon to the satellite can be expressed as [18]:

$$
a_T = -\mu_s (\frac{r - r_s}{||r - r_s||^3} + \frac{r_s}{r_s^3})
\tag{4}
$$

where $r$ and $r_s$ are the position vectors of the satellite and the sun or moon in the geocentric inertial system. $\mu_s$ is the gravitational coefficient.

- Solar radiation perturbation.

The perturbation acceleration caused by solar radiation on the satellite can be expressed as [18]:

$$a_R = KC_R \frac{S}{R} \frac{L_s}{4\pi c} \frac{r - r_s}{||r - r_s||^3} \tag{5}$$

where $C_R$ is the solar radiation coefficient. $c$ is the speed of light. $L_s$ represents the luminosity of the sun. $K$ is the solar visibility coefficient.

### 2.3.2. Satellite Model

The satellite model is composed of four subsystem models, namely, the GNC subsystem model, propulsion subsystem model, payload subsystem model and power subsystem model.

Among them, the GNC subsystem consists of a satellite body model, a solar wing sail model, a sensor model, a flywheel model and a controller model, as shown in Figure 4a, which is mainly responsible for the attitude and orbit control of the satellite. In order to maintain the satellite's payload to orient or track a specific target, attitude control is required. There are two ways to achieve attitude control. Small-scale attitude control is achieved by the controller generating a control signal to the flywheel, while large-scale attitude control is realized by the controller generating the switch signal of the attitude control engine.

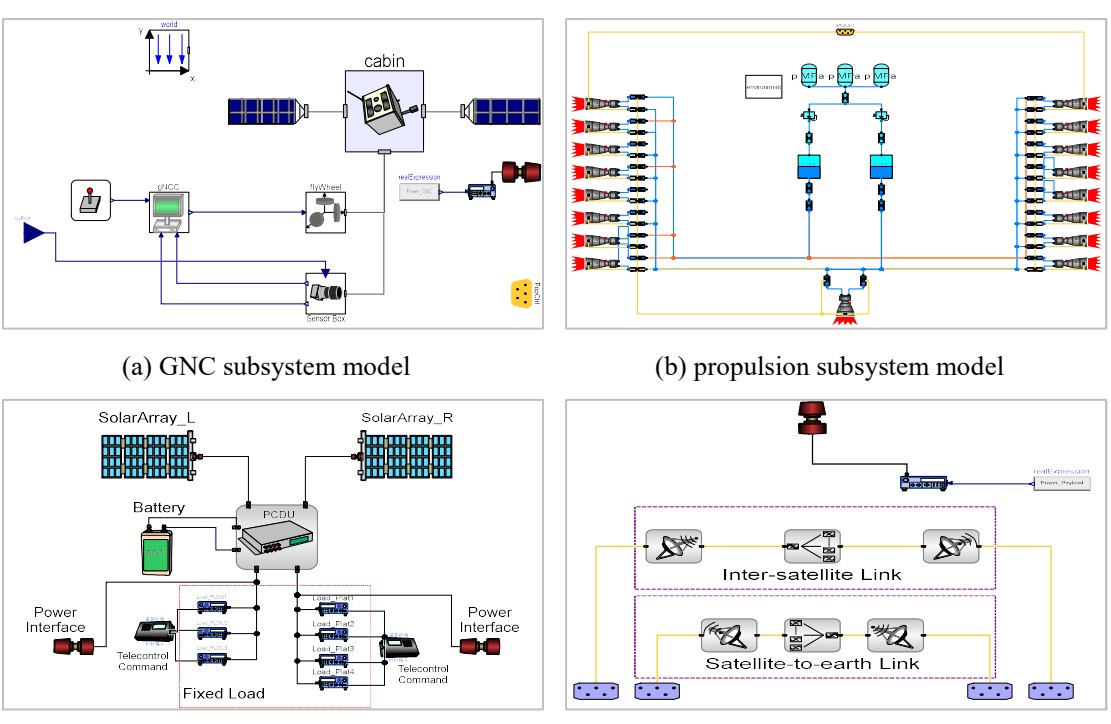

(a) GNC subsystem model

(b) propulsion subsystem model

(c) power subsystem model

(d) payload subsystem model

**Figure 4.** Four main subsystems that make up satellite model in MWorks.

The main task of the propulsion subsystem is to cooperate with the GNC subsystem to complete the attitude and orbit control during the life of the satellite. According to the function, it can be divided into a gas pressurization module, a propellant storage module and a thruster unit module, as shown in Figure 4b. The gas pressurization module is composed of three gas cylinder models, a gas orifice model and a pressure-regulating valve model. The main function is to provide the gas required for constant pressure operation to maintain working pressure for the rail-controlled engine; the propellant storage module

consists of two storage tank models and liquid orifice models. The main effect is to store, distribute and supply the propellant required by the engine; the unit module mainly includes one orbit control engine model and sixteen attitude control thruster models, which provide propulsion for orbit control and attitude adjustment.

The power subsystem, as shown in Figure 4c, includes two solar wing models, a battery model, a power controller model and load models. The solar wing models on the left and right sides are composed of a certain number of solar cell models combined in series and parallel, according to the design needs, and can generate full power under the condition of standard illumination of 1360 W/m$^2$. The battery model receives the electric energy output by the shunt regulator in the sunlit area for charging and discharges when the solar wing power is insufficient for the load to maintain the stability of the bus voltage. The main function of the power controller model is to realize the distribution and regulation of solar wing power generation, battery charging and discharging and load power and to shunt according to needs. The load model is divided into two types; one is the fixed power load given in Figure 4c, and the other is the load coupled in the other three subsystems. The second load is related to the working state of some equipment. For example, the power consumption of the electric load in the propulsion subsystem is related to the working status of seventeen engines.

The payload subsystem is generally composed of antenna models and transponder models, as shown in Figure 4d. For the LEO communication satellite constellation, it is mainly used to establish ISL [19] and satellite-to-earth links. The user's communication data is received by the receiving antenna of the corresponding frequency band and then sent to the transponder. After the transponder performs gain adjustment and signal amplification processing on the data signal, it is transmitted to the transmitting antenna and then to the next communication node through the transmitting antenna.

### 2.3.3. System Control Center Model

The system control center model belongs to the ground segment. The main function in the satellite constellation is to maintain the stability of the constellation structure. The input is the orbital factors of all satellites calculated by the constellation orbit model, and the output is the speed pulse that the GNC subsystem needs to generate the orbital maneuver control signal. In the satellite constellation structure, the two factors that have the greatest impact on the satellite coverage performance are the phase distribution and orbital plane distribution of the satellite, so phase control and orbital plane control are required.

The orbital plane and phase control reference of each satellite in the constellation are determined according to the following formula [20]:

$$\Omega_0^* = \sum_{j=0}^{p-1} \sum_{k=0}^{s-1} \lambda_{jk} (\Omega_{jk} - j \cdot \frac{2\pi}{P} + j \cdot \frac{2\pi}{P}) \tag{6}$$

$$u_{00}^* = \sum_{j=0}^{p-1} \sum_{k=0}^{s-1} \lambda_{jk} (u_{jk} - j \cdot F \cdot \frac{2\pi}{T} - k \cdot \frac{2\pi}{S})$$
$$+ j \cdot F \cdot \frac{2\pi}{T} + k \cdot F \cdot \frac{2\pi}{S} \tag{7}$$

where $\lambda_{jk}$ is the normalized weighting factor and the value is 1/T.

RAAN tolerance $\epsilon_\Omega$ and phase tolerance $\epsilon_u$ are given in the form of model parameters. When $|\Delta u + H_u \Delta \dot{u}| > \epsilon_u$, a phase-holding maneuver is required, and the required tangential velocity pulse $\Delta v_u$ is given by the following formula:

$$\Delta \dot{u}_r = \begin{cases} 0, & |\Delta u| < \varepsilon_u \\ K_u(\Delta u - \varepsilon_u), & \Delta u > \varepsilon_u \\ K_u(\Delta u + \varepsilon_u), & \Delta u < -\varepsilon_u \end{cases} \tag{8}$$

$$\Delta v_u = \frac{a}{3}(\Delta \dot{u} - \Delta \dot{u}_r) \tag{9}$$

where $\Delta \dot{u}_r$ is the phase deviation change rate correction value. $\Delta \dot{u}$ is the rate of change of the phase deviation, which is obtained by the difference. $K_u = -1/H_u$, and $H_u$ is the rate of change of the phase deviation, which is obtained by the difference.

Similarly, when $|\Delta \Omega + H_\Omega \Delta \dot{\Omega}| > \epsilon_\Omega$, the orbital maintenance maneuver is required, and the required normal velocity pulse $\Delta v_\Omega$ is given by the following formula:

$$\Delta i = -\frac{1}{3J_2 \sin i}(\frac{a}{R_E})^2 \sqrt{\frac{a^3}{\mu}}(\Delta \dot{\Omega} - \Delta \dot{\Omega}_r) \tag{10}$$

$$\Delta v_\Omega = 2\sqrt{\frac{\mu}{a}} \sin (i/2) \tag{11}$$

where the calculation of $\Delta \dot{\Omega}_r$ is similar to that of $\Delta \dot{u}_r$. $\Delta i$ is the required change in inclination. *J*2 refers to the perturbation of the earth's oblateness.

### 2.4. Satellite Constellation Model Validation

The verification of the correctness of the model is an indispensable link from the establishment to the application, and three methods are usually adopted: theoretical analysis, data comparison and simulation comparison. For equipment or systems that can obtain experimental data, validating and calibrating models with real experimental data must be the preferred method. If the device lacks experimental data but has a validated simulation model, the correctness of the model can be verified by comparing the simulation results generated under the same test scenarios. If neither of the above two are available, then the simulation results can only be qualitatively analyzed in combination with the principles of modeling. Figure 5 shows the calculation results of the orbit in MWorks and STK, including true anomaly, inertial position and subsatellite position. Obviously, the deviation between the calculation results of the constellation system model we established and the results of STK is far less than 5%, which can prove that the calculated results of our model in terms of orbit are credible.

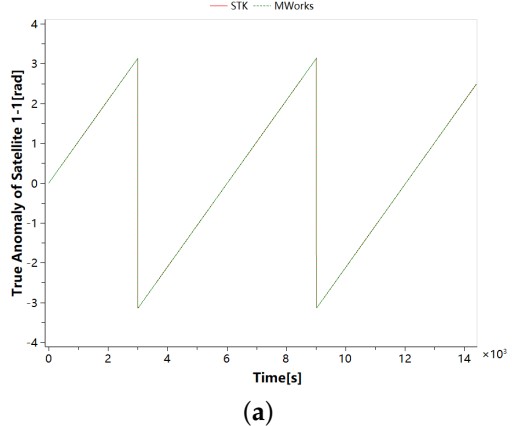

(**a**)

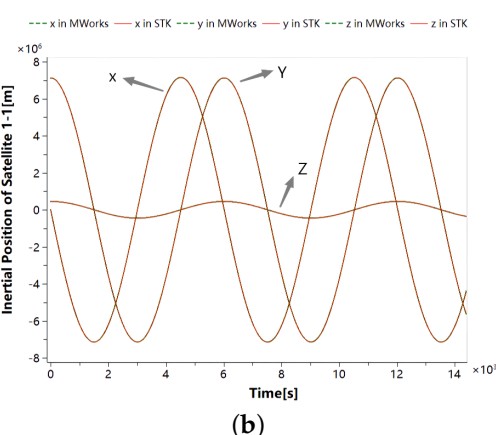

(**b**)

**Figure 5.** *Cont.*

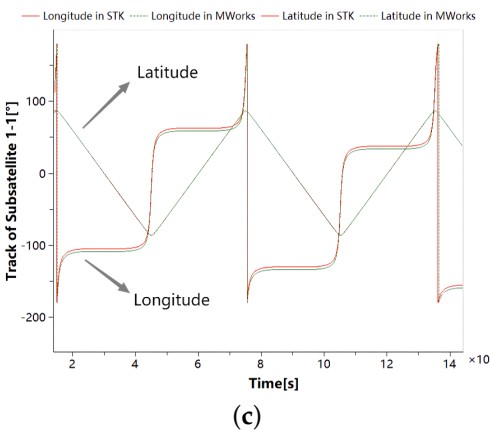

(**c**)

**Figure 5.** Comparison of MWorks Orbital Calculation Results with STK: (**a**) True Anomaly. (**b**) Inertial Position. (**c**) Subsatellite Point Track.

## 3. PHM System Driven by Model Data Hybrid

### 3.1. System Architecture

Referring to the design idea of the existing spacecraft health management system [21], the PHM system architecture proposed in this paper is shown in Figure 6, including the basic data layer and the health management layer. The basic data layer provides the data required to support health management, and the health management layer contains the health monitoring and fault diagnosis functions required to achieve on-orbit satellite constellation health management.

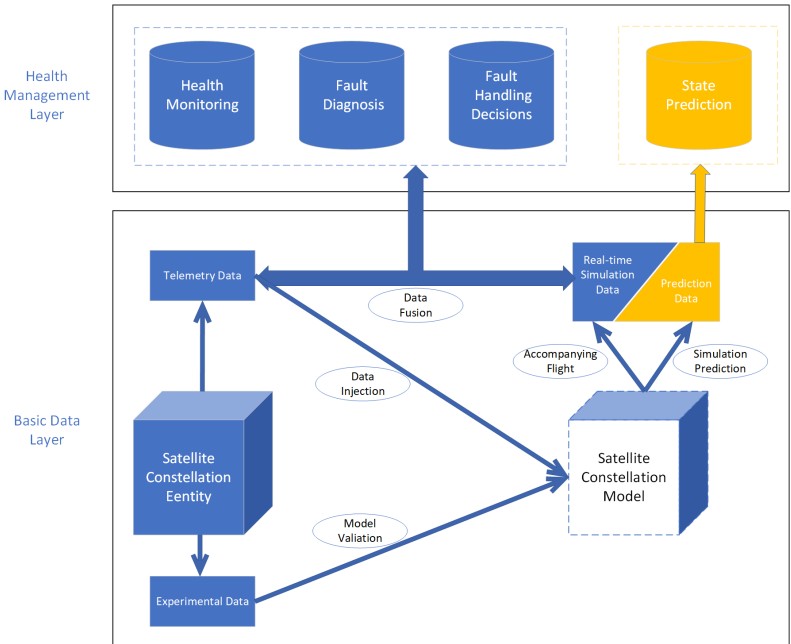

**Figure 6.** Health monitoring system architecture for satellite constellation.

### 3.1.1. Basic Data Layer

The main functions of the basic data layer are described as follows:

1.  Data injection. On the one hand, the constellation on-orbit telemetry data (including environmental parameters, orbital parameters and system parameters) are received through the telemetry data interface module and parsed into engineering values with practical physical meanings. On the other hand, command data that drive model state transitions are generated by the event schedule from mission planning. According to

the mapping relationship between the data and model variables, the telemetry data and command data are injected into the model to drive the simulation.

2.  Model synchronization. The injection of real-time telemetry data into the digital model can ensure that the status of the constellation in the digital space is consistent with the physical entity, so as to obtain high-fidelity and more comprehensive state data and support real-time state monitoring, fault diagnosis and other services.

3.  Simulation prediction. On the basis of model synchronization, another simulation thread is started to perform over-real-time simulation to obtain the subsequent state change in the constellation.

### 3.1.2. Health Management Layer

Based on the fusion of telemetry data and simulation data, the health management layer can provide health management services, which can be divided into four aspects: health monitoring, fault diagnosis, fault handling decision and status prediction.

1.  Health monitoring. The basis of the health management service is the real-time monitoring of the running status of the digital model of the satellite constellation. With the injection of real-time telemetry data, the digital model can track the dynamic changes of the physical entity. Then, simulation data, real data, and fused data between the two are extracted from digital models to fully reflect the running status of the constellation using visualization techniques. At the same time, the monitored data shall be interpreted in real time according to specific interpretation rules by means of threshold analysis and deviation analysis. If the data are judged to be abnormal, fault diagnosis shall be performed.

2.  Fault diagnosis. The fault diagnosis service is enabled when the health monitoring service has a fault warning. It analyzes the characteristics of the change process of abnormal telemetry parameters by combining abnormal telemetry parameters and the corresponding model simulation parameters. Then, using the simulation data as a reference, advanced fault identification algorithms are used to determine the faulty components and causes.

3.  Decision on fault handling. After the fault diagnosis service identifies the cause of the fault, the O&M personnel shall select a set of maintenance strategies from the prearranged maintenance plan and verify the effectiveness of such strategies on the digital model before implementation.

4.  Status prediction. Status prediction may occur in two scenarios: (1) Before executing a task, the O&M personnel obtain the running data of the next time period through status prediction and then evaluate the risk of the task based on the data and make predictive adjustments to the task plan. (2) The initial, inconspicuous fault features discovered by the health monitoring service may not be identified by the fault diagnosis algorithm. Through status prediction, more definite fault data can be obtained to support the fault diagnosis service.

## 4. Key Technology

### 4.1. Data Fusion

It is mentioned in the first section that pure data-driven PHM has three shortcomings, and data fusion can compensate for two of them. Data fusion is a technique that uses telemetry data to drive models to produce reliable simulation data, which is then supplemented by simulation data. The simulation data supplement the telemetry data in two dimensions. One is the time dimension. Due to the existence of areas without communication links, the telemetry data of satellites cannot be obtained for a long time in these areas during each orbital cycle. However, the simulation data produced by data fusion can fill the vacancy during this period, so as to realize the monitoring of satellites for the whole time period. Second is the quantity dimension. Due to the limitation of the transmission bandwidth and number of sensors, the telemetry data of the constellation are limited. It is likely that the variables measured by sensors are less sensitive to fault

characteristics. However, the simulation data do not have such restrictions. All variables involved in the model can be directly obtained. Therefore, data fusion can extend the input dimension of fault data to achieve more accurate fault detection and diagnosis.

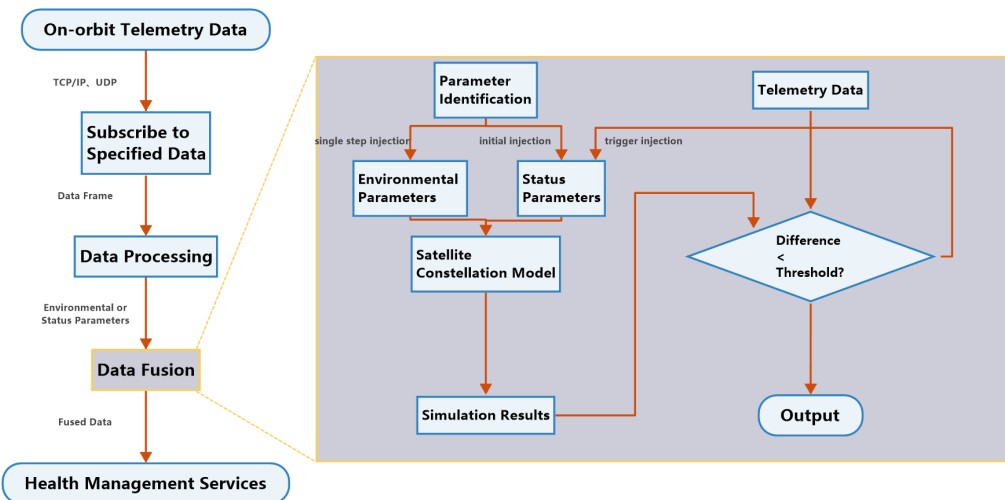

**Figure 7.** Flow chart of data fusion.

To achieve data fusion, it is necessary to build a bridge between the digital model and the physical entity and to establish a two-way mapping relationship between them. After the construction of the constellation, the ground control center is responsible for the direct monitoring and control of the constellation operation and is the data source of telemetry data. As shown in the left of Figure 7, a satellite in the constellation establishes a connection with the ground control center through TCP/IP or UDP protocol. It parses the source code of the subscription data frame and injects it into the digital model to drive the synchronization of models. Together with the telemetry data, the simulation data provide data support for health management services at the upper layer.

The process of data fusion is shown in the right of Figure 7. According to the characteristics of the data, the telemetry data are classified into two categories, namely, environmental parameters and status parameters. In order to ensure that the environment of the constellation model is consistent with that of the physical entity, environmental parameters, such as solar light intensity, solar incident angle and orbital information, need to be injected into the digital model at each simulation step, while for status parameters, such as equipment power consumption, bus voltage and current, temperature inside the device, attitude angle and attitude angle velocity of satellites, it is sufficient to inject them as the initial state in the first simulation step, and the subsequent simulation values are derived from the model principles. As the simulation time progresses, the error between the simulation data and telemetry data will gradually accumulate until the preset threshold is exceeded, the O&M personnel manually or the system automatically injects the telemetry values of the above state parameters into the model again in the area with communication to synchronize the state of the model and the physical entity and ensure the accuracy of the simulation data.

### 4.2. Health Monitoring

On the basis of simulation data and telemetry data and with the help of visualization tools, health monitoring shall not only comprehensively and fully display the operational status information of constellation to the O&M personnel but also perform a real-time evaluation of the constellation status to prevent failure.

For the polymorphic constellation states, the traditional monitoring method needs to define the basis for determining the health state of constellation. By refining or quantifying the health status of equipment, it is necessary to provide evaluation criteria for subsequent health status assessment. The health state division is classified in qualitative and quanti-

tative ways. In engineering applications, it is mostly classified quantitatively, that is, by determining the state boundaries of key status parameters, and the thresholds of each performance parameter need to be determined with sufficient expertise and the support of equipment test data. The abnormal condition detection is to determine the health status threshold in quantitative state division, and $M$ parameters are judged every $N$ seconds. When a status parameter exceeds the health threshold by more than or equal to $X$ times, an abnormal condition occurs. Generally, such a fault detection method refers to a parameter that exceeds a certain interval for several times continuously or a combination of "and" "or" for multiple judgment conditions. In more complicated cases, after one condition is satisfied, other conditions shall be transferred to the other conditions for further judgment.

Obviously, the traditional limit checking approach has unavoidable shortages. Firstly, there are a large number of abnormalities or symptoms that cannot be detected only by threshold checking. Secondly, the threshold value is highly dependent on domain knowledge, which is time-consuming and labor-intensive, and an unreasonable threshold value will result in either no anomalies being detected or a large number of false alarms, making the detection system insensitive to real anomalies. In order to address the limitations of the traditional methods mentioned above, various machine-learning-based fault detection methods, such as nonlinear regression, core element analysis and support vector machines, have been proposed. In particular, the one-class support vector machine (OCSVM) combined with the one-class classification (OCC) algorithm based on the support vector machine (SVM) has shown great potential in fault detection [22].

The basic principle of OCSVM is shown in Figure 8. The model uses the normal operation data of the constellation as training samples and maps it to the high-dimensional feature space through the kernel function. The operation data can be from historical data or credible simulation data [23]. Assuming the coordinate origin as the only abnormal sample point, an optimal hyper-plane exists to separate the training sample from the coordinate origin and maximizes their interval. The optimization objective can be expressed by Equation (12):

$$\min_{\omega, \xi, \rho} \frac{1}{2} \|\omega\|^2 + \frac{1}{vn} \sum_{i=1}^{n} \xi_i - \rho \tag{12}$$
$$subject\ to: \omega \cdot \phi(z_i) \geq \rho - \xi_i, \xi_i \geq 0$$

where $v$ is a user-defined parameter used to control the upper limit of the outliers number in the training set. Its value range is (0, 1). $\xi$ are the slack variables. $z_i$ is the element in the given data set. $n$ is the size of the data set. $\phi(z_i)$ stands for the mapping function corresponding to the kernel function. $\omega$ represents the vector perpendicular to the decision boundary. $\rho$ denotes the bias term.

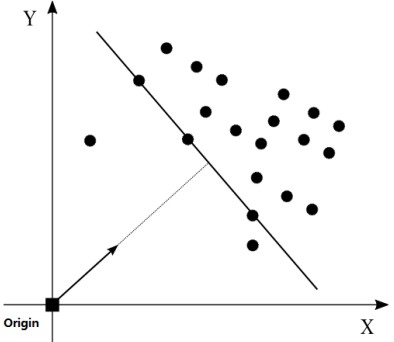

**Figure 8.** Principle diagram of OCSVM.

A RBF Gaussian kernel function is adopted, and its form is given as:

$$K(z_i, z_j) = exp(-\frac{\|z - z_j\|^2}{2\sigma^2}) \tag{13}$$

where $\sigma$ is the width parameter of the RBF Gaussian kernel.

By introducing the Lagrangian operator and selecting the Gaussian kernel function $K(z_i, z_j)$, the decision function of a new point can be expressed by Equation (14). When the decision function returns +1, it indicates that there is no anomaly, and if it returns $-1$, it means that the operation status data of the constellation are abnormal and need to be confirmed by engineers.

$$f(z) = sgn(\sum_{i=1}^{n} \alpha_i K(z_i, z_{new}) - \rho) \tag{14}$$

The calculation formula of $\rho$ is as follows:

$$\rho = \sum_i \alpha_i K(z_i, z_j) \tag{15}$$

### 4.3. Fault Diagnosis

When the health monitoring service issues a data abnormality warning, a possible fault of the constellation system requires an intervention of the fault diagnosis technique to confirm the cause of the fault. As a data-driven fault diagnosis technology, deep learning is used to extract fault characteristics from the training samples by learning from multiple hidden layers of network. Its strong characteristic learning and extraction capability can greatly improve the efficiency of fault diagnosis [24,25]. However, the success of deep learning in fault diagnosis requires a large number of labeled data for training. For satellite constellations, although there are many data sources, such as test data of individual equipment in the constellation system, data obtained during ground tests and telemetry data generated during operation in orbit, due to the design principles of the high reliability and high safety of the constellation, most of the satellites are operating under normal operating conditions and the high cost and long production period of the constellation, making it impossible to perform system-level fault tests. This will lead to abundant normal operation data and scarce failure data of the constellation, which are unfavorable for the training of neural networks.

In solving the problem of data imbalance, fault simulation technology based on system model would be of great value. The fault diagnosis workflow is shown in Figure 9. Based on the reliable system model described in Section 2.1, which has been corrected several times, after the fault modes are injected into the model as parameters, the model can generate the labeled fault data required for deep neural network training. However, the representation of fault modes in the model and the correlation between fault modes and model parameters still require a lot of preliminary work by designers. It is possible that not all fault modes can be expressed by the model. In addition, the more the fault modes are associated with model parameters, the more refined the granularity of model is, which will greatly reduce the efficiency of simulation. However, the time cost and economic cost are still quite advantageous compared with physical tests.

There are many deep learning algorithms used for fault diagnosis, such as auto-encoder (AE) [26], convolutional neural network (CNN) [27,28] and recurrent neural network (RNN) [29], which have their advantages and disadvantages. How to obtain high diagnosis rates in engineering applications with the aid of the above deep learning algorithms is the focus of the subsequent research.

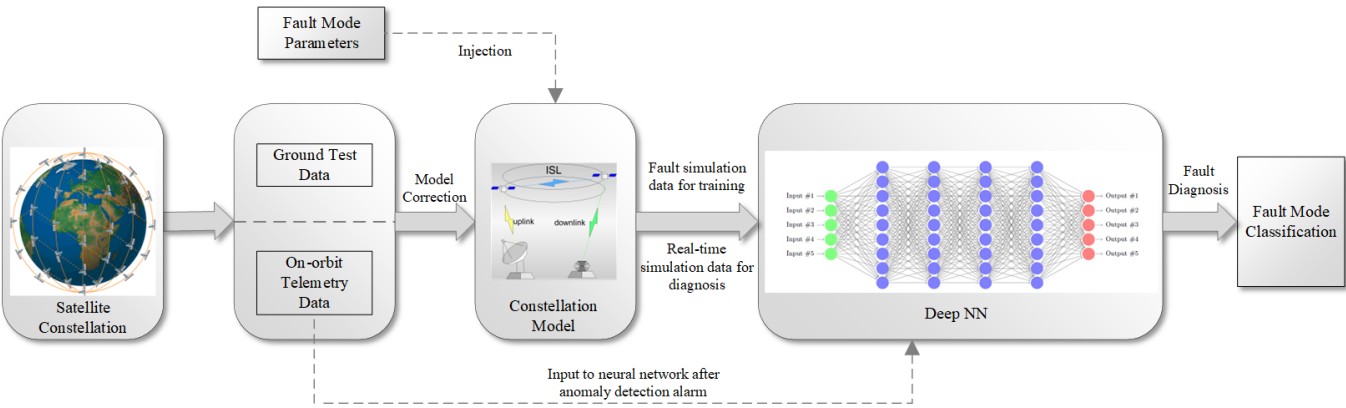

**Figure 9.** Fault Diagnosis Workflow.

### 4.4. Status Prediction

Status prediction technology is based on model synchronization and takes current state parameters as the starting point. Driven by the subsequent planned flight program, combined with the orbital and environmental parameters provided by the orbital computing module, over-real-time simulation is carried out to predict the status changes of the constellation in a certain period of time, as shown in Figure 10. The status prediction function has great value in the PHM system. The prediction data can not only be used as the input of the fault detection module to give advance warning of potential faults but can also be input to the fault diagnosis module for fault location to ensure that the O&M personnel have sufficient time to deal with possible faults.

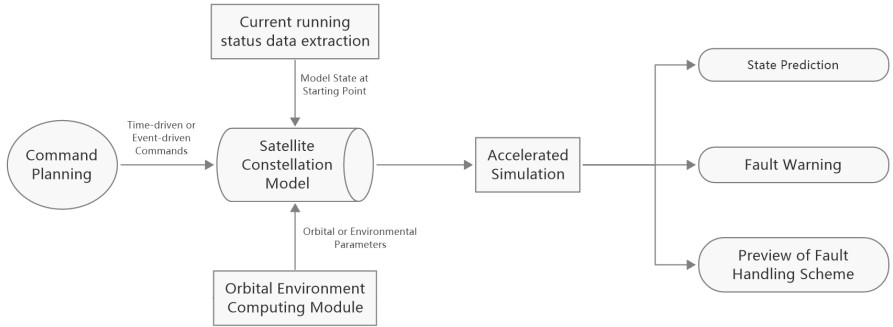

**Figure 10.** Realization process of state prediction.

### 4.5. Fault Handling Decision

During the design of the constellation, the designer will give full consideration to potential fault modes and make corresponding maintenance plans. After fault diagnosis service identifies the cause, a group of maintenance plans should be selected from the preplanned maintenance plans to implement the fault. Based on the digital model, the maintenance strategy can be injected into the model in the form of parameters or modules and simulated before implementation to verify its effectiveness. In addition, the O&M personnel can also adjust the maintenance strategy predictively with the aid of status prediction to achieve optimal O&M.

## 5. Case Study: A PHM System for Satellite Constellation

### 5.1. System Model and Parameters

As shown in Figure 11, the satellite constellation model is expressed in a unified form, supporting a multilevel and multidomain simulation. The satellite constellation library on the left contains orbital models, satellite model system control models, link models, etc. The right side is the system model integrated by the above models, which can provide a

multidimensional and multilevel view to show the different characteristics from satellite to link or from system to component.

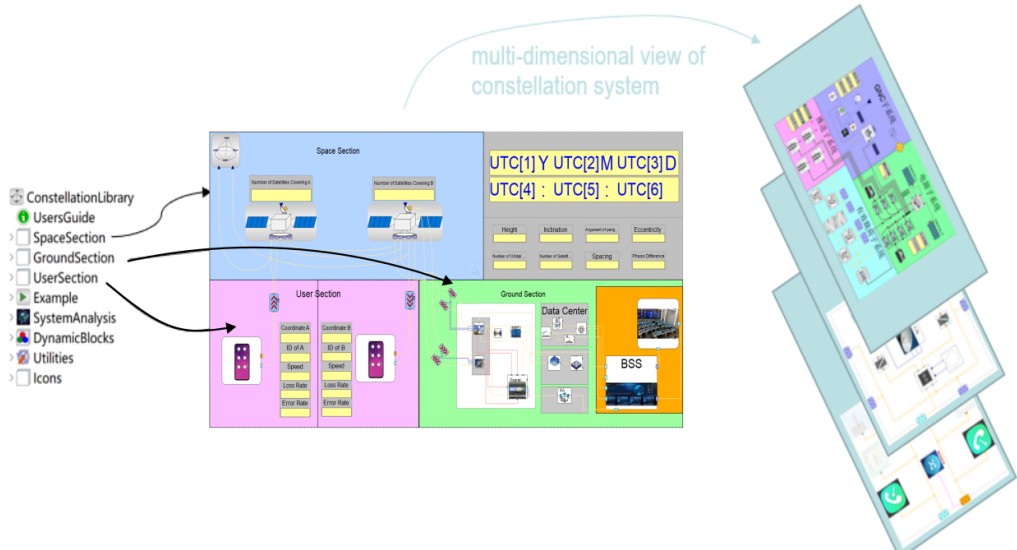

**Figure 11.** Example satellite constellation system model.

Based on the developed constellation model, users can simulate a dynamic response of a constellation system under reasonably defined constellation specifications, working conditions and operating parameters. In a constellation system with 20 satellites, when simulation duration and stepsize are set as 100 s and 0.1, system simulation actually consumes 842 s. The main parameters used in system simulation are listed in Table 2. Among them, constellation orbit parameters are from Iridium NEXT [30], and other satellite parameters are some reasonable defaults.

**Table 2.** Simulation parameters of the satellite constellation model.

| Parameter | Value |
| --- | --- |
| Number of orbital planes | 6 |
| Number of satellites per plane | 11 |
| Inclination | 86.4° |
| Eccentricity | 0.00126 |
| Argument of perigee | 90° |
| Ascending node | 31.6° |
| Mass of satellite | 900 Kg |
| Max flywheel torque | 32 N·m |
| Thrust of thruster | 5 N |
| Maximum power generation | 3200 W |
| Bus voltage | 28 V |

*5.2. Results of PHM System*

In this case, part of the results produced by the PHM system constructed based on the above framework are shown in Figure 12. The upper part shows the simulation results produced by the system at the satellite, orbit and link after injecting the data in Table 2. These results are fused with telemetry data as the input for system status monitoring. The status monitoring interface of PHM is shown in the lower left corner, which can provide a more intuitive and comprehensive display of the satellite constellation status. The curve in the lower right corner shows the prediction result of the PHM system based on the current satellite battery state of charge. Imagine that if the satellite needs to perform a high-power task after entering the shadow area, it is necessary for the O&M personnel to predict the

state of charge of the physical battery of the satellite in orbit during the mission planning stage. If the prediction results show that the power stored before the satellite enters the shadow area does not support the satellite to perform the mission, then the O&M personnel have sufficient time to adjust the mission plan.

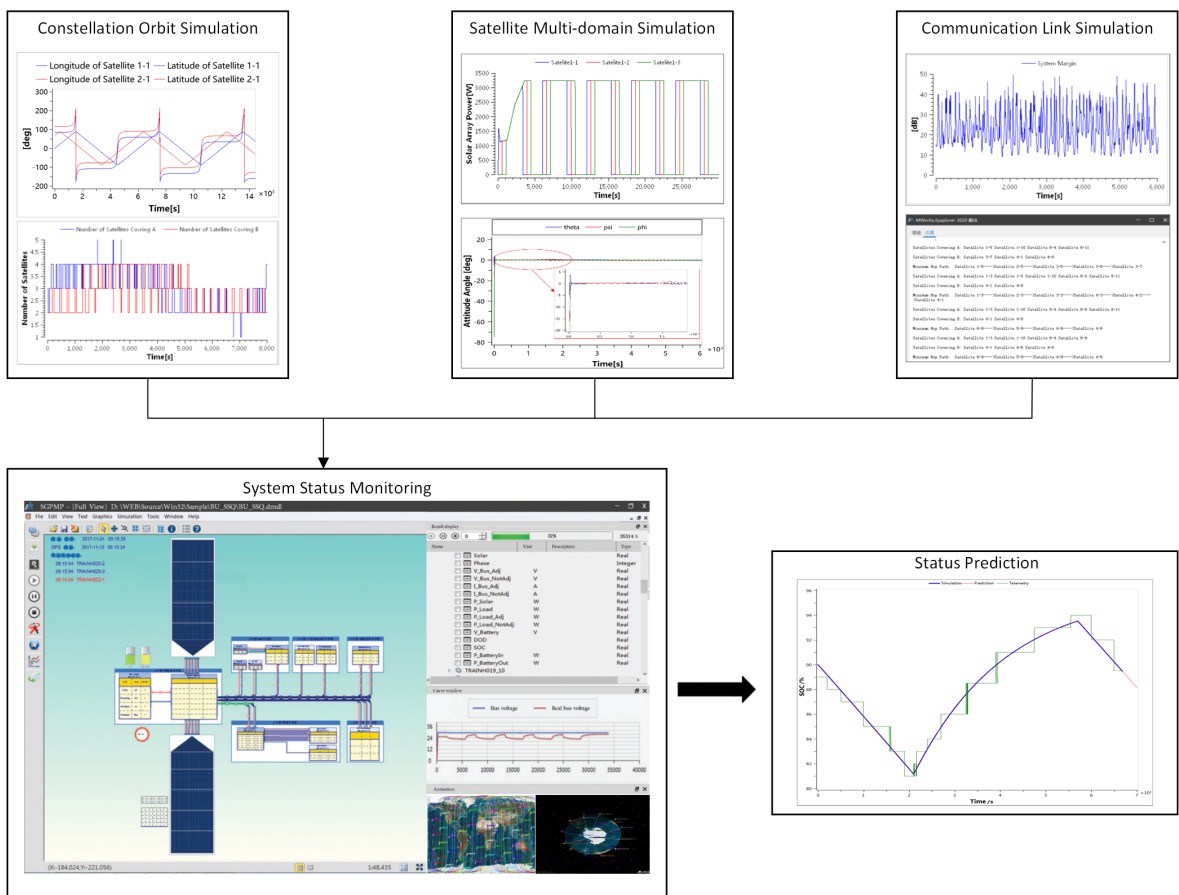

**Figure 12.** Simulation Results of PHM System.

## 6. Discussion

Due to the huge number of satellites in the LEO constellation and the changeable operating environment, the probability of system faults is much higher than that of a single satellite. Therefore, the design and operation stages at both ends of the constellation system life cycle are significant and challenging. Traditional spacecraft O&M often relies on logic defined early in the design process as well as real-time telemetry data. Through the case study in the previous section, the effectiveness of the framework proposed in this paper is proved, and the enlightenment for the design and O&M stage of the constellation system is as follows:

1.  The multidomain characteristics of the satellite constellation system can be expressed through a unified language. The constellation model expressed in the unified language is more flexible and convenient for the design and verification of the constellation system and is more conducive to the knowledge accumulation of the design unit.
2.  The model–data-hybrid-driven PHM method has more advantages than the traditional data-driven PHM method, and its advantages are reflected in the model's enhancement of data and the model's predictive function.
3.  The data enhancement of the model is manifested in two aspects. One is the expansion of telemetry data from the time dimension and the number of variables in state monitoring. The other is that the model can generate a large amount of labeled data for training fault diagnosis algorithms, as a supplement to the experimental data.

4. According to the existing research results, simulation prediction based on the mechanism model is more stable, reliable and generic than the data-based method.

However, there are still several issues to be addressed in future work.

1. The work efficiency of fault diagnosis, state monitoring or state prediction is closely related to the accuracy of the model. High-fidelity models help improve the performance of PHM systems. In this example, the orbital data and the calculation results of the satellite energy system are satisfactory, but there are still large errors in the calculation results of the information link, so the construction of a high-fidelity model of the satellite constellation system requires further in-depth research.
2. We hope to incorporate more efficient machine learning algorithms into the PHM framework proposed in this paper. Appropriate machine learning algorithms can play an active role in high-fidelity model construction, fault diagnosis and state prediction.
3. Due to the limitations of PC performance, we tested a constellation of up to 1000 satellites, and although the simulation worked, the efficiency was unacceptably low. In order to solve the problem of simulation, we plan to adopt the scheme of distributed simulation. Distributed simulation may not be a good choice for energy-based interfaces because it requires decoupling the variables coupled inside the energy-based interface into two causal-type interfaces, which is detrimental to the solution of the model. However, for the satellite constellation, there is no energy exchange between satellites, only information transfer, which is completely suitable for the distributed simulation scheme. We only need to arrange one master and dozens of high-performance slaves in a local area network to achieve real-time simulation.

## 7. Conclusions

Data-driven health management methods are widely used in the O&M of spacecraft, but there are still factors such as a limited number of sensors, a lack of fault data, and nonmeasurement and control areas that restrict the application of data-driven methods. In this paper, a satellite constellation model library is developed in MWorks and a PHM framework for satellite constellation health based on model–data fusion is proposed. The purpose is to use high-precision models to compensate for the deficiencies of pure data methods in various stages of testing, O&M. The work can be summarized as follows:

* A satellite constellation system model library has been developed in MWorks, which integrates the constellation orbit model, satellite multidomain model and communication link model. Based on this model library, a satellite constellation system can be built for ground coverage characteristic calculation and orbit control simulation, satellite energy balance simulation, link selection and margin calculation simulation.
* A PHM architecture for satellite constellation based on the Modelica model combined with a telemetry-data-driven model is proposed, which can also provide guidance for the design of a general spacecraft health management platform.
* Extracting the orbital parameters of the Iridium NEXT constellation and designing a verification example, it verifies the feasibility of building the constellation orbit model, satellite multidomain model and communication link model in a unified manner under the Modelica language and proves that the simulation model has a positive effect on the enhancement of telemetry data in the PHM of satellite constellation.

As for the outlook for further research, the focus will be on how to use a model–data hybrid for the training of fault diagnosis algorithms and how distributed solutions can improve the efficiency of satellite constellation system model simulation.

**Author Contributions:** Conceptualization, C.L., D.S. and L.C.; methodology, C.L. and L.C.; validation, C.L. and J.D.; writing—original draft preparation, C.L.; writing—review and editing, J.D.; visualization, C.L.; supervision, L.C. All authors have read and agreed to the published version of the manuscript.

**Funding:** This research was funded by National Key R&D Program of China (Grant No. 2019YFB1706501 00905) and Research on Model-Driven Design Method of Complex Product System (Grant No. 0270100065).

**Data Availability Statement:** Data supporting the findings of this study are available from the corresponding author upon request.

**Acknowledgments:** Thanks for the support of the members of the spacecraft digital escort team in Huazhong University of Science and Technology CAD Center and Suzhou Tongyuan Software and Control Tech. Co.

**Conflicts of Interest:** The authors declare no conflict of interest.

**Abbreviations**

The following abbreviations are used in this manuscript:

| | |
|---|---|
| PHM | Prognostics and Health Management |
| CPS | Cyber-Physical Systems |
| LEO | Low Earth Orbit |
| O&M | Operation and Maintenance |
| ISL | Inter-Satellite Link |
| RAAN | Right Ascension of Ascending Node |
| OCSVM | One-Class Support Vector Machine |
| SVM | Support Vector Machine |
| OCC | One-Class Classification |
| AE | Auto-Encoder |
| CNN | Convolutional Neural Network |
| RNN | Recurrent Neural Network |

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
