# Peer review of "Modeling of Satellite Constellation in Modelica and a PHM System Framework Driven by Model Data Hybrid"

_electronics, doi:10.3390/electronics11142155_

Round 1

Reviewer 1 Report

This paper firstly establishes the entire satellite constellation system model, which is integrated from the satellite multi-domain system, the constellation orbit environment system and the communication link system. Then, according to the technical concept of CPS, an implementation framework of a PHM system driven by model data hybrid for satellite constellation is proposed. The framework is based on model simulation data and telemetry data, and combines virtual and real data fusion, fault diagnosis, simulation prediction and other technologies to generate enhanced data to drive the effective operation of PHM system. Finally, a verification case is designed to prove that the satellite constellation health management system implemented under this framework has a positive effect on the reliable operation and maintenance of the satellite constellation system.

The paper idea is valid and adds knowledge to the literature. The paper is related to the journal scope. Overall, the paper can be accepted after a major corrections. 

The limitation of the paper lies in paper structures and its presentation as it has to be improved.

·       The related works are missing and a new recently published paper has to be cited, e.g:

o   Al-Hourani, A. (2021). An analytic approach for modeling the coverage performance of dense satellite networks. IEEE Wireless Communications Letters10(4), 897-901.

o   Palanisamy, S.; Thangaraju, B.; Khalaf, O.I.; Alotaibi, Y.; Alghamdi, S. Design and Synthesis of Multi-Mode Bandpass Filter for Wireless Applications. Electronics 2021, 10, 2853. https://doi.org/10.3390/electronics10222853

o   Okati, N., Riihonen, T., Korpi, D., Angervuori, I., & Wichman, R. (2020). Downlink coverage and rate analysis of low Earth orbit satellite constellations using stochastic geometry. IEEE Transactions on Communications68(8), 5120-5134.

o   Dai, C. Q., Zhang, M., Li, C., Zhao, J., & Chen, Q. (2020). QoE-aware intelligent satellite constellation design in satellite internet of things. IEEE Internet of Things Journal8(6), 4855-4867.

o   Palanisamy, S., Thangaraju, B., Khalaf, O. I., Alotaibi, Y., Alghamdi, S., & Alassery, F. (2021). A Novel Approach of Design and Analysis of a Hexagonal Fractal Antenna Array (HFAA) for Next-Generation Wireless Communication. Energies, 14(19), 6204.

o   Osoro, O. B., & Oughton, E. J. (2021). A techno-economic framework for satellite networks applied to low earth orbit constellations: Assessing Starlink, OneWeb and Kuiper. IEEE Access9, 141611-141625.

·       The discussion is missing and it has to be added before conclusion.  

·       The paper has to be proofread.

Author Response

Point 1: The limitation of the paper lies in paper structures and its presentation as it has to be improved.

Response 1: First of all, thank you for your patience in reading our paper and giving so many professional opinions. We think this arrangement of thesis structure is more appropriate at present. We invited native English speakers to improve the presentation of our paper.

Point 2: The related works are missing and a new recently published paper has to be cited.

Response 2: We added citation to paper in section 2.3.1, numbered[16].

Point 3: The discussion is missing and it has to be added before conclusion. 

Response 3: A new section on the discussion has been added to this paper.

Point 4:   The paper has to be proofread.

Response 4: We have proofread this paper again.

Reviewer 2 Report

It was an interesting paper. It presented a simulation modeling for satellite constellation and framework for prognostics and health management. 

The paper stated that "For a constellation system with 20 satellites, when 448 simulation duration and stepsize are set as 100s and 0.1, system simulation actually 449 consumes 842s".

For the Mega Constellations such as the Starlink and OneWeb with many thousands or tens thousands of satellite, the model may be able to be in real-time.

The Case studies of Iridium Next was not very good examples, as it only had about 66 satellites.

It would also be important to compare with the SOA software tools mentioned in the paper to be able to deal with thousands and even many 10 thousands of satellite.

From the paper, it was not clear how to check if such complicated system would be able operate correctly.

In addition, some abbreviations such as PHM, GNC, ..., etc. should be spelled out when used first time in the texts. Line colours was not easy to read when printed in black and white. References may be needed for equations such as (1) - (6). 

Therefore, the main revision should be made improve the quality of the paper.

Author Response

Point 1: The paper stated that "For a constellation system with 20 satellites, when 448 simulation duration and stepsize are set as 100s and 0.1, system simulation actually 449 consumes 842s".For the Mega Constellations such as the Starlink and OneWeb with many thousands or tens thousands of satellite, the model may be able to be in real-time.The Case studies of Iridium Next was not very good examples, as it only had about 66 satellites.

Response 1: First of all, thank you for your patience in reading our paper and giving so many professional opinions. All simulation and verification work is carried out on the PC. We have tested 1,000 satellites. The model can be simulated and solved, but the time it takes to complete one orbital cycle is unacceptable, so we use the Iridium Next Constellation for testing.

Point 2: It would also be important to compare with the SOA software tools mentioned in the paper to be able to deal with thousands and even many 10 thousands of satellite.

Response 2: We use STK software a lot, so the simulation results of STK are used as a reference in this paper. Like you, we have concerns about the viability of the technical framework proposed in this paper in constellations of 1000+ satellites. We feel that the focus of this paper is on the preliminary verification of the feasibility of the technical framework. In the future, we will consider the introduction of distributed simulation technology and model order reduction technology to solve the simulation efficiency problem of large-scale satellite constellations.

Point 3: From the paper, it was not clear how to check if such complicated system would be able operate correctly.  

Response 3: The correctness of the system is mainly reflected from two aspects, the model and the algorithm. The verification method for model correctness is described in Section 2.4. The correctness verification of the algorithm (taking machine learning as an example) is achieved by evaluating the performance of the trained neural network on the training set, and there are also mature methods, but this verification method is not the focus of the PHM architecture, so this article does not describe.

Point 4: In addition, some abbreviations such as PHM, GNC, ..., etc. should be spelled out when used first time in the texts. Line colours was not easy to read when printed in black and white. References may be needed for equations such as (1) - (6).

Response 4: We have revised and improved the problems you pointed out in the paper.

Round 2

Reviewer 1 Report

The authors have considered all my commands so it can be accepted in current form. 

Author Response

Point 1: The authors have considered all my commands so it can be accepted in current form. 

Response 1: We sincerely thank you for your professional opinion in the review.

Kind regards.

Reviewer 2 Report

The authors only considered about 1,000 satellites. The Iridium is relative older system. Given the Starlink and OneWeb and other mega-constellations, the authors should really consider a few thousands or a few tens of thousands of satellites with results. 

Otherwise, the works presented in the paper would not have much value to the subject areas.

At least, the authors should discuss how to deal with the mega-constellations.

Author Response

Point 1: The authors only considered about 1,000 satellites. The Iridium is relative older system. Given the Starlink and OneWeb and other mega-constellations, the authors should really consider a few thousands or a few tens of thousands of satellites with results. 

Otherwise, the works presented in the paper would not have much value to the subject areas.

At least, the authors should discuss how to deal with the mega-constellations.

Response 1: We sincerely thank you for your professional opinion in the review.

           We understand your concerns about the applicability of our proposed method in the  mega-constellations. Our current equipment is limited to 1000 satellites and currently does not have the ability to test larger constellations. We plan to use distributed simulation to solve the simulation problem of mega-constellations, and we will respond to your concerns about this part in the discussion.

Round 3

Reviewer 2 Report

The author has addressed all my queries.

I do not have any further comments.